# Toward New Epidemiological Landscapes of *Trypanosoma cruzi* (Kinetoplastida, Trypanosomatidae) Transmission under Future Human-Modified Land Cover and Climatic Change in Mexico

**DOI:** 10.3390/tropicalmed7090221

**Published:** 2022-09-02

**Authors:** Constantino González-Salazar, Anny K. Meneses-Mosquera, Alejandra Aguirre-Peña, Karla Paola J. Fernández-Castel, Christopher R. Stephens, Alma Mendoza-Ponce, Julián A. Velasco, Oscar Calderón-Bustamante, Francisco Estrada

**Affiliations:** 1Instituto de Ciencias de la Atmósfera y Cambio Climático, Universidad Nacional Autónoma de México, Ciudad de Mexico 04510, Mexico; 2C3—Centro de Ciencias de la Complejidad, Universidad Nacional Autónoma de México, Ciudad de Mexico 04510, Mexico; 3Instituto de Ciencias Nucleares, Universidad Nacional Autónoma de México, Ciudad de Mexico 04510, Mexico; 4Institute for Environmental Studies, VU Amsterdam, De Boelelaan 1087, 1081 HV Amsterdam, The Netherlands; 5Programa de Investigación en Cambio Climático, Universidad Nacional Autónoma de México, Ciudad de Mexico 04510, Mexico

**Keywords:** chagas disease, global change, spatial datamining, global warming, neglected disease

## Abstract

Chagas disease, caused by the protozoa *Trypanosoma cruzi*, is an important yet neglected disease that represents a severe public health problem in the Americas. Although the alteration of natural habitats and climate change can favor the establishment of new transmission cycles for *T. cruzi,* the compound effect of human-modified landscapes and current climate change on the transmission dynamics of *T. cruzi* has until now received little attention. A better understanding of the relationship between these factors and *T. cruzi* presence is an important step towards finding ways to mitigate the future impact of this disease on human communities. Here, we assess how wild and domestic cycles of *T. cruzi* transmission are related to human-modified landscapes and climate conditions (LUCC-CC). Using a Bayesian datamining framework, we measured the correlations among the presence of *T. cruzi* transmission cycles (sylvatic, rural, and urban) and historical land use, land cover, and climate for the period 1985 to 2012. We then estimated the potential range changes of *T. cruzi* transmission cycles under future land-use and -cover change and climate change scenarios for 2050 and 2070 time-horizons, with respect to “green” (RCP 2.6), “business-as-usual” (RCP 4.5), and “worst-case” (RCP 8.5) scenarios, and four general circulation models. Our results show how sylvatic and domestic transmission cycles could have historically interacted through the potential exchange of wild triatomines (insect vectors of *T. cruzi*) and mammals carrying *T. cruzi*, due to the proximity of human settlements (urban and rural) to natural habitats. However, *T. cruzi* transmission cycles in recent times (i.e., 2011) have undergone a domiciliation process where several triatomines have colonized and adapted to human dwellings and domestic species (e.g., dogs and cats) that can be the main blood sources for these triatomines. Accordingly, Chagas disease could become an emerging health problem in urban areas. Projecting potential future range shifts of *T. cruzi* transmission cycles under LUCC-CC scenarios we found for RCP 2.6 no expansion of favourable conditions for the presence of *T. cruzi* transmission cycles. However, for RCP 4.5 and 8.5, a significant range expansion of *T. cruzi* could be expected. We conclude that if sustainable goals are reached by appropriate changes in socio-economic and development policies we can expect no increase in suitable habitats for *T. cruzi* transmission cycles.

## 1. Introduction

Chagas disease is widespread across the Americas and represents an important public health problem. The World Health Organization (WHO) has estimated that approximately 25 million people are at risk of infection, and between 6 and 7 million people are infected [1,2]. It has been estimated that around 5.5 million people in Mexico in particular are potentially affected by Chagas disease, with approximately 69,000 new infections annually [3,4]. However, there are no official programs for vector control and only a passive national surveillance program to monitor vector-borne transmission, which comprises over 96% of infection incidence in Mexico [3,5,6].

The etiological agent of Chagas disease is the protozoan *Trypanosoma cruzi* (Chagas,1909) (Kinetoplastida, Trypanosomatidae), which is transmitted by the infected feces of triatomine bugs upon entering the human bloodstream [3,7]. In natural conditions the life cycle of *T. cruzi* alternates between insect vectors (triatomine bugs of the family *Reduviidea*, subfamily *Triatominae*) and vertebrate hosts (mainly mammal species) [5,8]. Currently, 40 species of triatome bugs belonging to the genera *Rhodinus* Stål, 1859, *Triatoma* Laporte, 1832 and *Panstrongylus* Berg, 1879 [9,10] have been reported in North America as being naturally infected by *T. cruzi*. In Mexico, 21 of these species have been confirmed as positive for *T. cruzi*, and therefore could act as potential vectors of Chagas disease [11,12,13]. To date, 37 wild mammal species, belonging to the orders Marsupiala, Edentata, Chiroptera, Carnivora, Arthiodactyla, Rodentia and Primates, have been confirmed positive for *T. cruzi* [8,13,14,15,16,17]. However, this number could potentially represent a gross underestimation due to the possible existence of multiple unknown mammal hosts [8]. Additionally, various domestic mammals have been confirmed positive for *T. cruzi* (e.g., dogs, cats, and rats) [1,4,6,13,16], thereby increasing the potential exposure of human populations to the pathogen. This diversity of vectors and vertebrate hosts reflects the complexity of pathogen transmission cycles that can occur in sylvatic and domestic environments [18].

Transmission of *T. cruzi* has evolved from an enzootic disease of wild species [15,19] to an anthropozoonosis, which occurs when humans become infected contact with wild triatomine species carrying the pathogen [2,15,20]. Human contact with infected species can occur when humans move into natural habitats or when wild animals or vectors enter human dwellings [2,14,21]. There are three principal types of transmission cycle for *T. cruzi*; (1) a domestic cycle, which is maintained by humans, domestic animals, and triatomine bugs adapted to human dwellings [4,6,10,19]; (2) a wild transmission cycle, which involves wild mammals such as rodents or marsupials, and wild populations of triatomines [10,14]; (3) a peridomestic transmission cycle, which can be considered as a bridge between the other two transmission cycles. This transmission cycle originated in the wild transmission cycle and maintains infection among domestic animals in areas surrounding human dwellings through the action of peridomestic triatomines, and occasionally through exchanges with the wild transmission cycle (for example, dogs, and cats hunting wild animals, or wild animals entering areas near to human dwellings) [2,16,19].

The domestic transmission cycle is regarded as the most important cycle in maintaining the circulation of the pathogen in human populations, and has been responsible for concentrating the burden of disease in human communities [22]. However, until now, there have been few evaluations of the contribution of peridomestic and wild transmission cycles to the incidence of *T. cruzi* [14,16,23,24]. Although wild transmission cycles have been considered less important in generating human infections [10], current environmental changes (e.g., deforestation, agricultural and livestock expansion) have increased the probability of contact between humans and wild animals carrying the pathogen (vector and hosts) and, consequently, altering the dynamics of *T. cruzi* transmissions.

Commonly, the risk of *T. cruzi* infection has been related to poverty and has been classified as a primarily rural disease [25]. However, the rapid and disorganized growth of urban areas, which involves the invasion of wild species’ habitats, can substantially alter the frequency of parasite–vector–host–human interactions, generating a dynamic that can significantly increase the risk of *T. cruzi* infection in urban areas [26]. Recently, triatomines infected by *T. cruzi* have been collected in major cities [27,28]. These findings indicate that domestic mammals (dogs, cats) and synanthrope species such as rats could play an important role as blood sources for triatomine bugs in the absence of sylvatic mammals [4,6,26]. Certainly, an important factor that determines which vectors can invade and colonize human settlements is the natural landscape surrounding these urban and peri-urban locations. Accordingly, the incidence of *T. cruzi* infections has probably been underestimated in urban areas and may in the future become a more acute public health problem in densely populated cities.

There is a growing concern about how human-modified landscapes can drive emergence of Chagas disease due to changes in land cover and reduction of wild diversity [14,19]. However, as mentioned, little attention has been given to the compound effect of human-modified landscapes and current climate change on the transmission dynamics of *T. cruzi*. Climate change can modify the dynamics of vector-borne diseases through different mechanisms, including through the pathogen itself, through vector and host distributions, and through changes in transmission routes. For instance, high temperatures can speed up the development of *T. cruzi* and increase the number of infective forms (e.g., metacyclic and bloodstream trypomastigote) in vectors [29,30]. In warmer temperatures, insects may develop shorter life cycles with more than one generation per year, which can lead to higher population densities [30,31]. In addition, high temperatures combined with low humidity cause hematophagous insects to increase their feeding rates [32]. Climate change has the potential to alter or extend the natural ranges of vectors, hosts, and pathogens and to convert some regions that were previously uninhabitable into suitable habitats [33,34]. 

In this paper, we assess how wild and domestic transmission cycles are related to climate conditions and human-modified landscapes. By using a datamining framework [35,36], we measured correlations between the presence of *T. cruzi* transmission cycles, land use, land cover, and climate. We then estimated the potential range changes of *T. cruzi* transmission cycles under future land-use and land-cover changes and climate change scenarios (LUCC-CC) for 2050 and 2070 time-horizons. Finally, we quantified the potential range expansion or contraction of three types of *T. cruzi* transmission cycles, i.e., sylvatic, rural, and urban. 

## 2. Materials and Methods

### 2.1. Data of Wild and Domestic Cycles of T. cruzi

To assess the temporal and spatial distribution patterns of *T. cruzi* transmission cycles, we used as proxies geographic information for wild and domestic mammals considered as hosts, and data for those triatomine insects positive for *T. cruzi*, collected in domestic and sylvatic habitats. This information was obtained from the Atlas of Infectious Diseases of Mexico project (PINCC-UNAM), which is the first Mexican repository of etiological agents (vectors and hosts) of several zoonosis distributed in Mexico. We considered only those records that included geographic information (latitude–longitude), date, and habitat where the specimens were collected and tested for *T. cruzi* infection. Based on the habitat where the species were collected, they were grouped into sylvatic, rural, and urban transmission cycles.

### 2.2. Land-Used/Cover-Change Datasets

To evaluate the ecological relationships and potential temporal changes of *T. cruzi* transmission cycles, we used land-use/cover-change scenarios (LUCC) recently developed for Mexico [37,38]. The land-cover classes consisted of eight natural covers, i.e., cloud forest, grassland, hydrophilic vegetation, scrubland, temperate forest, tropical evergreen forest, tropical dry forest, and other vegetation types; and four anthropogenic covers, i.e., pastures, irrigated agriculture, rainfed agriculture, and urban. LUCC scenarios contained four historical (1985, 1993, 2002, 2011) and two future (2050, 2070) time horizons. 

Future LUCC projections considered three scenarios that integrate climatic data based on the representative concentration pathways (RCPs), demographic and economic growth linked to the shared socioeconomic pathways (SSPs), and deforestation and regeneration trends of natural cover. These were (1) the green scenario, which uses RCP 2.6, SSP1, the lowest deforestation rates and the highest regeneration rates of natural cover; (2) the business-as-usual (BAU) scenario, which uses RCP 4.5, the assumptions of SSP2 for demographic and economic growth, and mean historical deforestation and regeneration rates; (3) the worst-case scenario, which utilizes RCP 8.5 data, SSP3 assumptions, and the highest deforestation rates and lowest regeneration rates of natural cover. The LUCC projections were built using four general circulation models (GCM) (CNRMC-M5; GFDL-CM3; HADGEM2-E5; MPI-ESM-LR) [37,38].

### 2.3. Climate Datasets

We generated two bioclimatic variables (Bio1—annual mean temperature and Bio12—annual precipitation) for different time horizons in the past, using monthly temperature and precipitation values from 1969–2018 available at the Worldclim portal. We generated Bio1 and Bio12 variables using monthly variables averaged for the following time horizons: (1) 1976–1995 (1985), (2) 1984–2003 (1993), (3) 1993–2012 (2002), and (4) 2002–2018 (2011) to combine climate and LUCC scenarios. Additionally, we obtained future scenarios for the same variables for two time horizons: (1) 2041–2060 (2050) and (2) 2061–2089 (2070). For future climatic projections we selected three RCPs (2.6, 4.5, and 8.5) and the same four GCMs used in the LUCC projections [37,38].

### 2.4. Modelling Framework

#### 2.4.1. Characterizing the Epidemiological Landscapes of *T. cruzi* Transmission 

We used a Bayesian spatial datamining framework [35] to quantify the relationships among *T. cruzi* transmission cycles, LUCC, and climate conditions. This methodology allowed us to build a conditional probability model [*P* (*C_i_*|*X_k_*)] of the geographic distribution of a given target class, *C_i_*, (e.g., a rural *T. cruzi* transmission cycle), conditioned on ***X***, where ***X*** represents the presence or not of one or more predictive factors (e.g., land cover or temperature values) associated with a given spatial region. To quantify and determine which co-distribution, *P* (*C_i_*|*X_k_*), showed a statistically significant correlation for a given niche factor *X_k_* ∈ ***X***, we applied a binomial test, epsilon [*ε* (*C_i_*|*X_k_*)], which measured the statistical dependence of *C_i_* on *X_k_* relative to the null hypothesis, *P* (*C_i_*), that the distribution of *C_i_* is independent of *X_k_* and randomly distributed over the selected area [35,39]. Values of |*ε*| > 1.96 correspond to a greater than 95% confidence that *P* (*C_i_*|*X_k_*) occurs at a rate inconsistent with the null hypothesis, in the case that the binomial distribution can be approximated by a normal distribution. As *ε* increases monotonically, it can be used as a suitable measure of the relative importance of each niche factor, *X_k_*, for the presence or not of our target class, *C*. For example, in the case where *X_k_* represents an urban cover or a temperature value (e.g., 25 °C), *ε* is a measure of its potential importance for the presence of an urban cycle of *T. cruzi*. By using ε we ranked the land covers, temperature, and precipitation ranges to evaluate which combination of these factors was important for the establishment of urban, rural, and sylvatic cycles of *T. cruzi* transmission.

To obtain *ε* values for each *T. cruzi* transmission cycle and each environmental factor, we sampled our region of study (the continental region of Mexico) with a uniform grid of rectangular cells (*x_α_*) of 5 km × 5 km. We overlapped each geographical dataset (*T. cruzi* transmission cycle records, LUCC, and climatic layers) to assign their respective presence in each spatial cell, *x_α_*. We then counted the number of cells occupied for each factor, and the co-occurrences of *T. cruzi* transmission cycles with each LUCC class and each temperature and precipitation range. Finally, we calculated *ε* (*C_i_*|*X_k_*) for each cycle–variable pair combination: *ε* (*𝙲_T. cruzi_*|*𝚇_LUCC_*), *ε* (*𝙲_T. cruzi_*|*𝚇_temp_*), *ε* (*𝙲_T. cruzi_*|*𝚇_prec_*), …, *ε* (*𝙲_T. cruzi_*|*𝚇_n_*). To identify changes in the relationships of *T. cruzi* transmission cycles with LUCC and climatic variables, we first grouped *T. cruzi* transmission cycle occurrences in each time horizon (1985, 1993, 2002, 2011) in order to link them with corresponding historical climate and LUCC layers. Thus, for each historical time-horizon we obtained *ε* values for each *T. cruzi* transmission cycle and each environmental factor.

#### 2.4.2. Geographic Models of Potential Distribution of *T. cruzi* Cycles

We mapped the potential distribution of *T. cruzi* cycles by using the Bayesian *score* function *S* (*C*|*X_(xα)_*) [35]. This function provides a measure of the probability of finding the presence of *C* (presence of a given *T. cruzi* transmission cycle) when the ecological profile in a spatial region is *X_(xα)_*, where ***X*** is the full set of variables used. We quantified the score contribution, *S*(*C*|*X_k_*), of each individual LUCC class and each temperature and precipitation range (*X_k_*), to the presence of domestic or sylvatic *T. cruzi* transmission cycles. We then applied the resulting model to any spatial cell (*x_α_*) by adding the individual contributions in that cell, that is, *S* (*C*|*X_(xα)_*) = *S* (*X_land cover_*) + *S* (*X_temp_*) + *S* (*X_prec_*) + … + *S* (*X_n_*); thus, we determined the ecological profile, ***X***, for each spatial cell (*x_α_*). Higher or lower value of *S* (*C*|*X_(xα)_*) indicates favorable or unfavorable conditions, respectively, for the presence of *C* [35].

We calculated the score function for the 1985 historical time horizon as the baseline LUCC-CC model for each type of *T. cruzi* transmission cycle. To test the 1985 LUCC-CC models for sylvatic, rural, and urban *T. cruzi* transmission cycles, we randomly split the presence data of each *T. cruzi* cycle, using 70% to train the models and 30% to validate them. We calculated the area under the curve (AUC) statistic of the receiver operating characteristic (ROC) plot to evaluate models’ performance. Models with AUC > 0.70 are thought to be useful, and models are considered good to excellent if AUC > 0.90. We then applied this model to each historical (1993, 2002, 2011) time-horizon. Models for 1993, 2002, and 2011 were evaluated by calculating AUC values using 100% of the *T. cruzi* data corresponding to each time period. We measured the changes (loss or gain) of range area for each type of *T. cruzi* transmission cycle, to obtain the percentage of area lost or gained from 1985 to the more recent time horizons (1993, 2002, and 2012). We then applied the historical LUCC-CC model (i.e., the 1985 model) to future LUCC and climatic scenarios for each RCP and GCM. The four CGMs were consolidated for each RCP scenario (green, BUA, and worst-case) and for each year (2050, 2070) by averaging score values. Finally, we calculated the range differences between the historical LUCC-CC model and the future projections for green, BAU, and worst-case scenarios [37,38]. These differences represent the projected increasing or decreasing range shifts of potential *T. cruzi* cycle presence. Finally, for each year and RCP scenario, we calculated the net change of range shift, which was the difference between the area with suitable conditions gained and suitable area lost in future (i.e., percentage of area gained – percentage of area lost). Positive values of net change indicated that area gained was greater than area lost, and therefore a potential risk of emergence of the pathogen in new regions.

## 3. Results

We obtained information for 21 triatomine, 52 wild mammal, and 12 domestic mammal species, confirmed as positive for *T. cruzi* (Appendix A). 67% (14 out of 21) of the triatomine species were collected in domestic habitats, thereby highlighting the domiciliation process of a great number of bugs (e.g., *Triatoma dimidiata* (Latreille, 1811), *Triatoma pallidipennis* (Pinto, 1927)). Only 27% (14 out of 52) of the wild mammal species were collected exclusively in natural habitats, which means that >70% of the mammals have been observed in both natural and anthropogenic habitats. Records of domestic species that are positive for *T. cruzi* include domestic pets (dogs and cats), livestock species (e.g., cows, horses, pigs), and synanthropic rodents (rats and mice). Dogs and synanthropic rodents have been recorded in both wild and domestic habitats; therefore, these species could be an important link between domestic and wild transmission cycles.

By correlating *T. cruzi* transmission cycle presence (i.e., vector and/or host presence) with LUCC and climate, we characterized those environmental conditions (i.e., eco-epidemiological landscapes) that might favor pathogen transmission. We identified the individual and combined contributions of anthropogenic and natural vegetation covers to *T. cruzi* presence. We found a significant relationship between sylvatic transmission cycles and tropical dry forest (ε > 3), as well as a significant relationship (ε > 1.96) with agricultural and urban covers for the periods 1985 and 1993 (Figure 1), thereby indicating that anthropogenic landscapes near to tropical dry forest have a high probability of peridomestic transmission cycle presence through potential interaction between wild, synanthropic, and domestic species. For 2011, sylvatic transmission cycles had a significant probability of being established near grazing areas (ε = 2.1). Consequently, livestock species may have played an important role in maintaining *T. cruzi* transmission cycles.

For rural transmission cycles we observed a significant correlation with agricultural land (ε > 1.96) as well as urban cover, tropical dry forest, and tropical evergreen forest in 1985, 1993, and 2002 time horizons (Figure 1), thus highlighting the importance of natural habitats that surround anthropogenic landscapes promulgating the interaction between domestic and wild hosts and vectors. However, we noted that only agriculture and urban covers had significant ε values in 2011, indicating that domestic and synanthropic species were the main species maintaining *T. cruzi* transmission cycles. Similarly, this was observed for urban transmission cycles; in 2002 and 2011, the presence of *T. cruzi* was mainly related to urban areas (ε > 10). Apparently, there has been a domiciliation process of triatomine species in urban populations, where domestic species are the main bloodmeals of triatomine bugs.

For climatic conditions, we observed temporal changes in relationships between *T. cruzi* presence and temperature and precipitation conditions. Urban transmission cycles showed an increase in temperature ranges from the earliest to most recent time horizons. In 1985 and 1993, temperatures ranged between 17.8 and 24.3 °C. However, more recently, *T. cruzi* has been distributed in places with temperatures > 25 °C. On the other hand, in 1985 and 1993, rural transmission cycles were distributed in sites with temperatures ≥ 22.9 °C; in contrast, suitable temperatures in 2011 were >24 °C. For sylvatic transmission cycles, suitable temperatures ranged between 20 and 25 °C (Figure 2). Regarding precipitation, domestic transmission cycles (urban and rural) had a significant relationship with ranges > 1000 mm. In 2011, sylvatic transmission cycles were distributed in places with precipitation between 400 and 700 mm (Figure 3).

By mapping the score contribution for LUCC-CC, we identified those regions with suitable conditions for the establishment of *T. cruzi* transmission cycles. Distribution models of *T. cruzi* cycles (sylvatic, rural, and urban) for our baseline model for 1985 and projections to 1993, 2002, and 2011 showed a high level of accuracy, with AUC values ranging from 0.7 to 0.98 (Appendix A). According to our baseline LUCC-CC model for 1985, we observed that sylvatic transmission cycles were mainly distributed in the Pacific and Gulf of Mexico slopes, and this distribution was maintained into recent time horizons (2002 and 2011; red areas in Figure 4), but with an increase from 19 to 36% of geographic area with suitable conditions in 1993, 2002, and 2011 (Figure 5). Conversely, urban and rural transmission cycles were widespread across both the slopes and in the center of Mexico (red areas; Figure 4), and their ranges increased from 7 to 17% into recent times (1993 to 2011; Figure 5). Models for the potential future distribution of suitable conditions for *T. cruzi* transmission cycles showed an insignificant range increase for all types of transmission cycles in the green scenario, where the net change values for all transmission cycles were <0 (i.e., range gained less than range lost; Figure 6). Accordingly, changes in socio-economic development policies that reduce deforestation could prevent an expansion of favorable conditions for the presence of the pathogen. In contrast, the BAU and the worst-case scenarios showed significant range increases, from 20 to 50% for urban and rural transmission cycles and more than 100% for sylvatic transmission cycles (Figure 5), with a positive net change (i.e., range gained greater than range lost; Figure 6). Therefore, LUCC and climate changes may favor the increase of sites with favorable conditions for the establishment of *T. cruzi* transmission cycles. Figure 7 shows that for 2050 and 2070, the geographic distribution of *T. cruzi* could expand towards the center of Mexico in the BAU and the worst-case scenarios. Unfortunately, the most densely populated Mexican states (e.g., Mexico City and the State of Mexico) lie in this region.

## 4. Discussion

Chagas disease, caused by the protozoan *T. cruzi*, is an important yet neglected disease, which affects a great number of people in Mexico. Until recently it has been considered to be related to poverty and rural populations. However, due to the invasion of natural areas, land-use processes, land-cover change, and climate change, the disease may spread to different non-endemic regions, including urban areas. Previous studies have evaluated the influence of human-modified land cover at the local level, for triatomines and for mammals infected by *T. cruzi* [14,27,28,40]. Studies addressing the influence of climate change on Chagas disease risk have focused on triatomine species distribution at the regional level, but without including information for other species that are positive for *T. cruzi* [41] and are therefore potential agents in the transmission cycle. Consequently, no link has been formally established between triatomine distribution and *T. cruzi* presence, nor any association with the influence of changes in land cover and climate [42,43]. Our study is the first attempt to assess the combined effect of human-modified land cover and climate change at a regional scale on the historical and potential future distribution of *T. cruzi*, using information that includes vector and mammal species confirmed as positive for the pathogen as proxies for domestic and sylvatic transmission cycles.

By correlating *T. cruzi* presence with historical LUCC and climate conditions, we characterized the eco-epidemiological landscape that has favored *T. cruzi* presence. Our results show how sylvatic and domestic transmission cycles could have interacted through the potential exchange between wild triatomines and mammals carrying *T. cruzi*, due to the proximity of human settlements (urban and rural) to natural habitats (e.g., tropical dry forest) [16,44]; mainly over the time horizons 1985 and 1993. However, in more recent times (i.e., 2011) *T. cruzi* transmission cycles have undergone a domiciliation process, where several triatomines have colonized and adapted to human dwellings. Additionally, domestic species such as pets (e.g., dogs and cats) and livestock species (e.g., cows and horses) could be the main blood sources for these triatomines and consequently become hosts of *T. cruzi*. Therefore, *T. cruzi* transmission cycles in rural and urban areas will no longer depend on the arrival of wild species to human settlements, since domestic species could maintain transmission cycles [45,46]. Accordingly, Chagas disease could become an emerging health problem in urban areas in the short and medium term [26].

Assessing the role of climatic conditions in the distribution of *T. cruzi* transmission cycles, we found that warmer temperatures were related to *T. cruzi* presence in recent time-horizons (1993, 2011). Temperatures play an important role in pathogen development and vector capacity. Optimal temperatures of *T. cruzi* development occur between 23 °C and 28 °C [29]. In recent times (1993, 2011), domestic cycles were present in regions with temperatures > 22 °C, which offered suitable conditions for *T. cruzi*. In addition, activity, density, and biting rates of insects increase significantly with temperature [30,31,47]. Therefore, warmer temperatures can expose humans to larger populations of triatomines and to more frequent interactions. 

To understand the environmental changes that could be implicated in the emergence of Chagas disease, we should consider the composite effect of several factors. In our study, by using land-use-land-cover-change–climate-change models, we included two principal factors that could potentially be related to Chagas disease emergence. On the one hand, land-use change processes, such as deforestation and fragmentation, can increase infestation rates of triatomes in domestic areas [19,48], and on the other hand, warmer temperatures can have a direct effect on the biology of *T. cruzi* and triatomine vectors. Our ecological and geographic models were able to identify combinations of these factors that could favor the establishment of *T. cruzi* transmission cycles. Thus, by assessing the historical influence of environmental changes on *T. cruzi* transmission-cycle distributions, we found that the loss of natural habitats (e.g., tropical dry forest) near to domestic environments could be associated with an increase in the domestic presence of bugs, thereby favoring the expansion of domestic transmission cycles from natural habitats to urban zones, and that temperature increases could expand the range of *T. cruzi* from tropical regions (Mexico slopes) to temperate zones (such as the center of Mexico).

Projecting potential trends in future environmental conditions under different LUCC and RCP scenarios, we evaluated potential range shifts of *T. cruzi* transmission cycles. Changes in temperature and rainfall conditions have the potential to influence the geographic distribution of *T. cruzi* by allowing more habitats to become suitable for its vectors and host species, and consequently promoting the establishment of new cycles of transmission in non-endemic areas [43]. However, as mentioned above, we should also consider the role of human-modified land cover in the distribution of *T. cruzi*. Under the green scenario, it is expected that changes in socio-economic development policies that reduce deforestation could prevent an expansion of favorable conditions for the presence of *T. cruzi* cycles. However, if the same socio-economic development policies continue (BAU scenario) or are exacerbated by growing demand and overexploitation of resources (worst-case scenario), a significant range expansion of *T. cruzi* could be expected. In summary, human-modified land cover and climate change threaten to transform Chagas disease from a rural and local problem to a national and urban one.

Chagas disease was one of the relatively neglected tropical diseases that were targeted for control and elimination by the 2020 London Declaration goals [49]. However, it continues to be a serious public health problem. As Chagas disease was included for elimination as a public health problem in the 2021–2030 road map set by the WHO in the 2030 Agenda for Sustainable Development [50], to reach this goal we need a better understanding of the complexity of *T. cruzi* transmission, particularly to achieve the interruption of transmission through the vectorial route. We should consider the full variety of *T. cruzi* transmission cycles (sylvatic and domestic) and how environmental change (land cover and climate change) can influence their establishment. According to the green scenario, if we reach sustainable goals by appropriate changes in socio-economic development policies, we can expect a scenario where there is no net increase in suitable habitats for *T. cruzi* transmission cycles.

## Figures and Tables

**Figure 1 tropicalmed-07-00221-f001:**
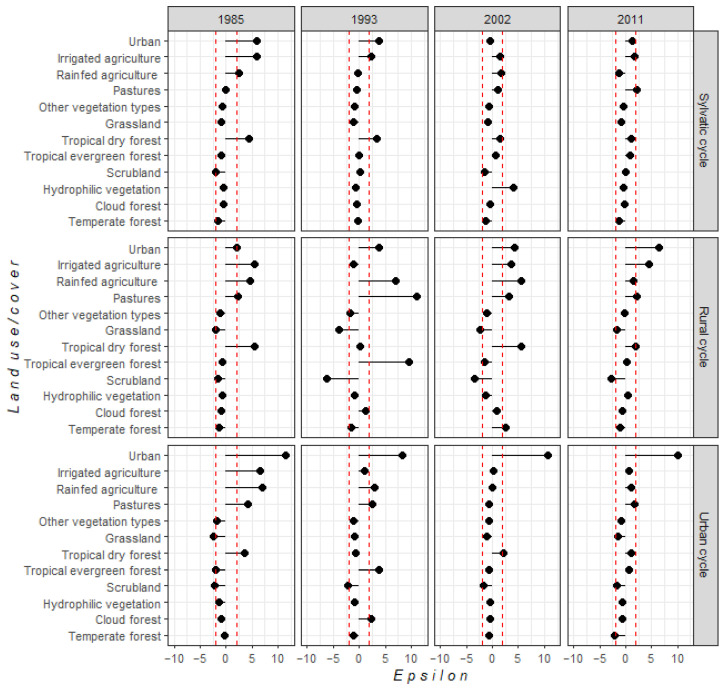
Statistical relationship between each land-use/cover class and each *T. cruzi* cycle, based on ε-values. The red dotted line indicates confidence intervals of statistically significant correlation (i.e., |ε| > 1.96).

**Figure 2 tropicalmed-07-00221-f002:**
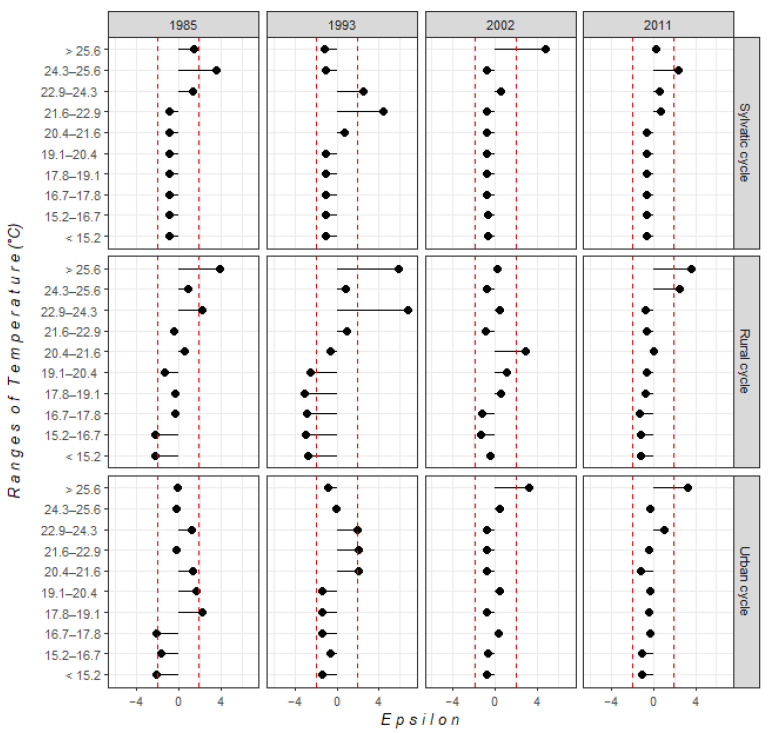
Statistical relationship between each temperature range and each *T. cruzi* transmission cycle, based on ε-values. The red dotted line indicates confidence intervals of statistically significant correlations (i.e., |ε| > 1.96).

**Figure 3 tropicalmed-07-00221-f003:**
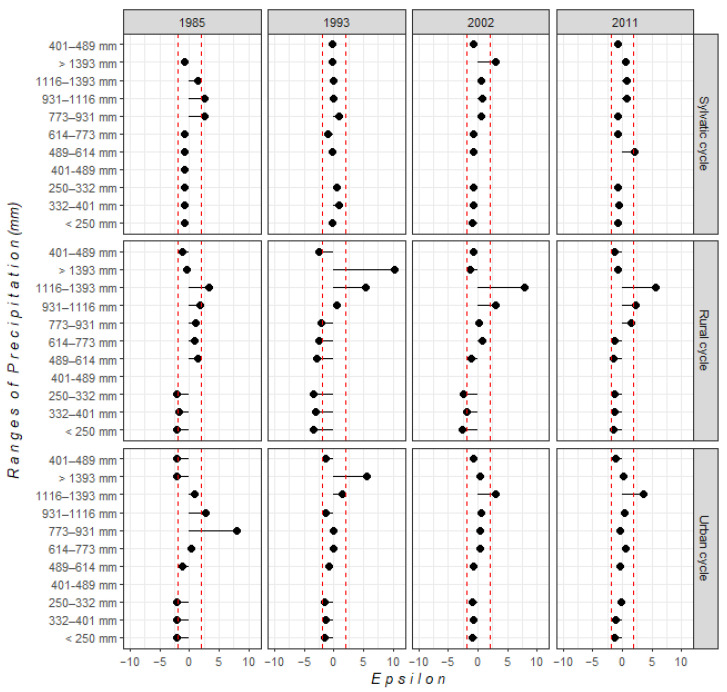
Statistical relationship between each precipitation range and each *T. cruzi* transmission cycle, based on ε-values. The red dotted line indicates confidence intervals of statistically significant correlations (i.e., |ε| > 1.96).

**Figure 4 tropicalmed-07-00221-f004:**
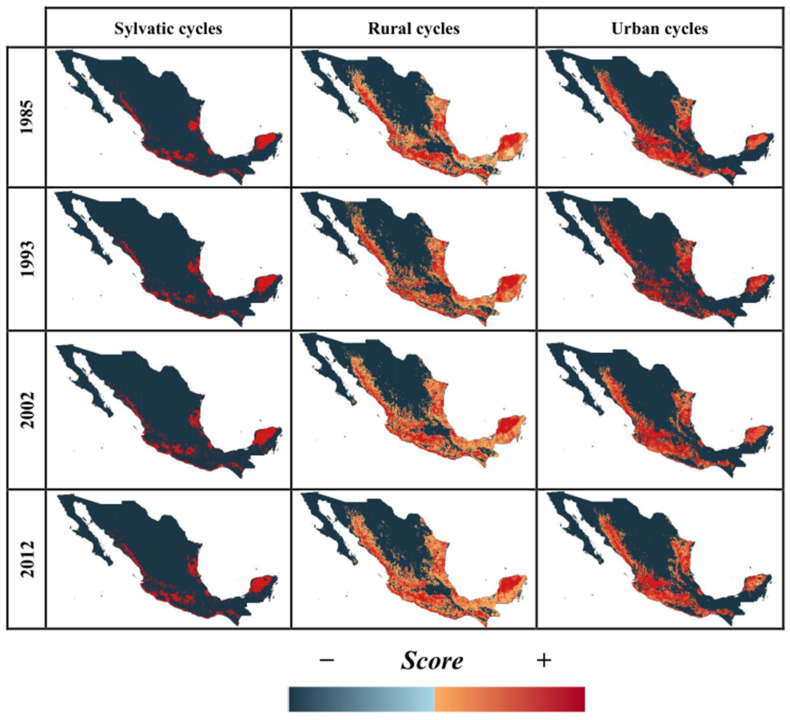
Potential distribution of each type of *T. cruzi* transmission cycle under historical conditions of land-use and land-cover-change and climate. The change from blue to red indicates a gradient from unsuitable (negative score) to suitable (positive score) ecological conditions.

**Figure 5 tropicalmed-07-00221-f005:**
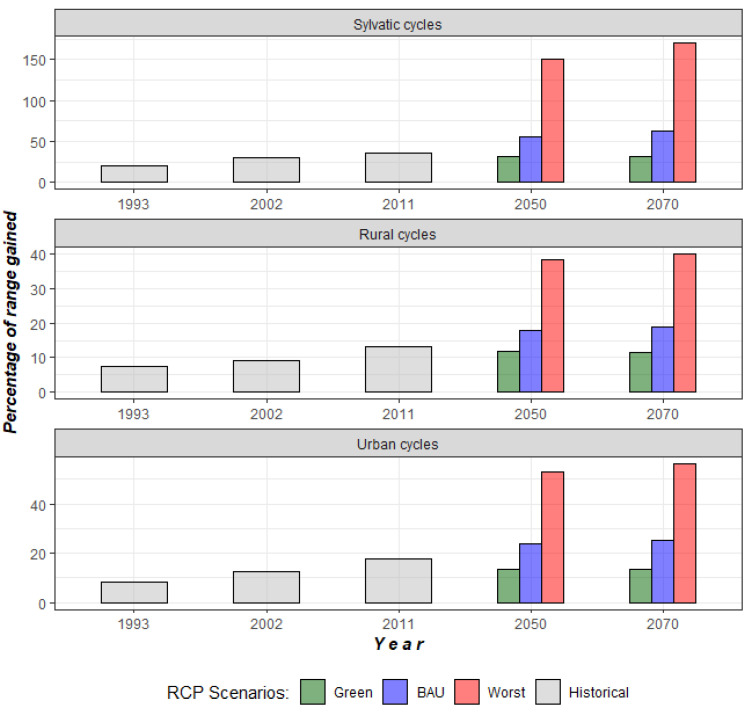
Percentage of range gained from historical LUCC-CC model (1985) to recent time-horizons (1993–2011) and for future distribution projections to 2050 and 2070, in each RCP scenario.

**Figure 6 tropicalmed-07-00221-f006:**
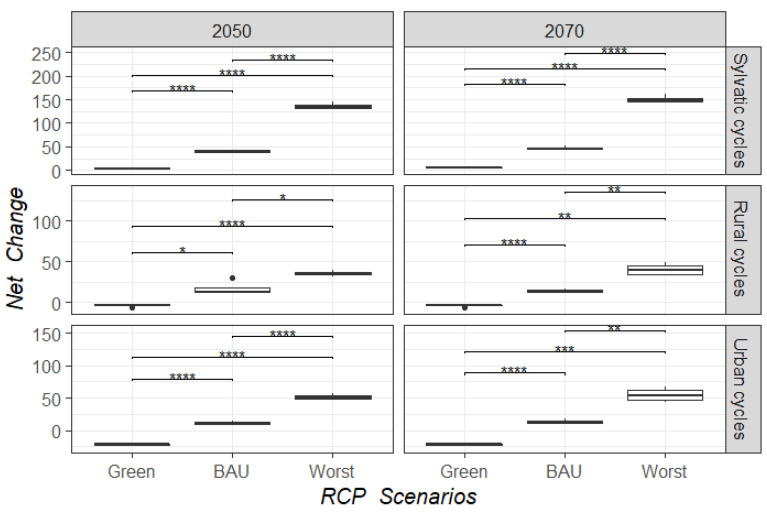
Net change of range shift for each type of *T. cruzi* transmission cycle under future LUCC-CC scenarios. The significance levels (*p*-values) for the differences between the average values of net change of the four GCMs in each RCP is indicated by asterisk(s) above the boxes: **** *p* <0.0001, *** *p* <0.001, ** *p* <0.01, * *p* <0.05, based on *t* testing.

**Figure 7 tropicalmed-07-00221-f007:**
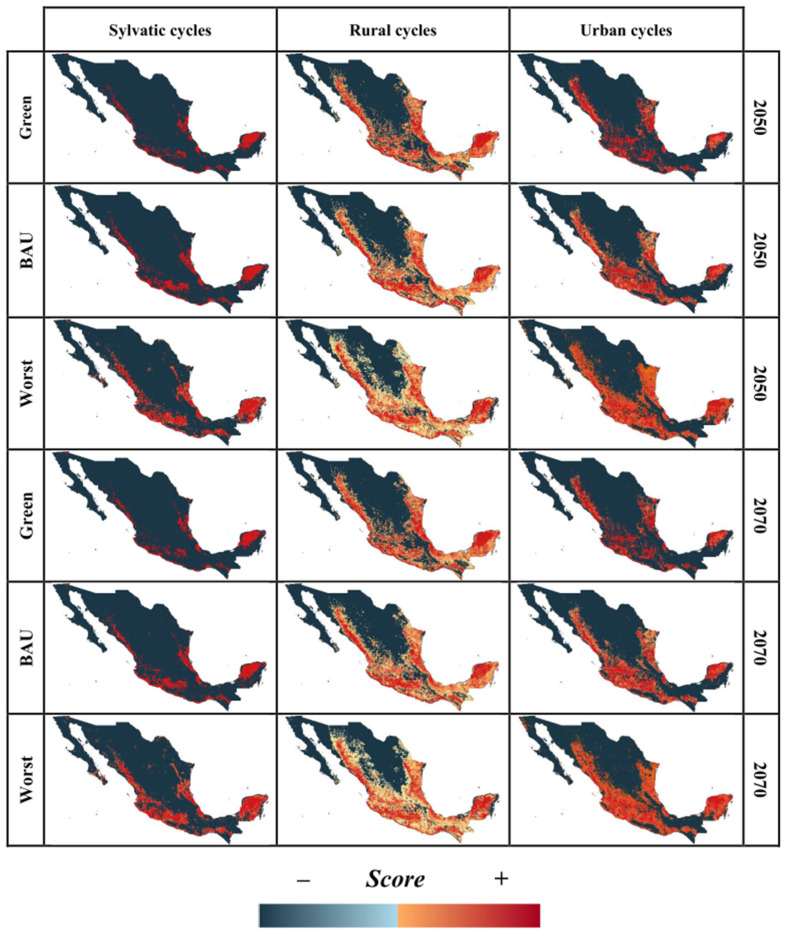
Predicted distribution of *T. cruzi* transmission cycles under future LUCC-CC scenarios ensembled for each year and each RCP scenario. The change from blue to red indicates a gradient from unsuitable (negative score) to suitable (positive score) ecological conditions.

## Data Availability

Not applicable.

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
