# Peer review of "Toward New Epidemiological Landscapes of Trypanosoma cruzi (Kinetoplastida, Trypanosomatidae) Transmission under Future Human-Modified Land Cover and Climatic Change in Mexico"

_tropicalmed, 2022, doi:10.3390/tropicalmed7090221_

Round 1
Reviewer 1 Report
In this work González-Salazar et al., used public available data of Triatomine bugs distribution in Mexico coupled to projections of climate change and land use to predict the epidemiological transmission of T. cruzi. For that, temperature changes, humidity and the use of the land, as well, the kind of T. cruzi life cycle was considered and the authors found that in recent times T. cruzi cycles have undergone a domiciliation processes, with several triatomines have adapted to human dwellings and that domestic species, such as dogs and cats, became the main blood sources for these triatomines, which in the end will no longer need the migration of wild tritomine species to the urban regions to maintain the transmission of T. cruzi. Moreover, the authors found that increases in deforestation, associated with warmer temperatures favors the domestic cycle of the parasite. Also, the warmer temperatures will increase the presence of Triatomine in temperate zones of Mexico. Altogether, the authors showed the impact of climate changes in the distribution of triatomines and how this could affect the epidemiology of Chagas Disease in Mexico.
Thus, the results presented in this work are important and will contribute to the parasitology community.
Author Response
Response to Reviewer 1 Comments
Point 1: In this work González-Salazar et al., used public available data of Triatomine bugs distribution in Mexico coupled to projections of climate change and land use to predict the epidemiological transmission of T. cruzi. For that, temperature changes, humidity and the use of the land, as well, the kind of T. cruzi life cycle was considered and the authors found that in recent times T. cruzi cycles have undergone a domiciliation processes, with several triatomines have adapted to human dwellings and that domestic species, such as dogs and cats, became the main blood sources for these triatomines, which in the end will no longer need the migration of wild tritomine species to the urban regions to maintain the transmission of T. cruzi. Moreover, the authors found that increases in deforestation, associated with warmer temperatures favors the domestic cycle of the parasite. Also, the warmer temperatures will increase the presence of Triatomine in temperate zones of Mexico. Altogether, the authors showed the impact of climate changes in the distribution of triatomines and how this could affect the epidemiology of Chagas Disease in Mexico.
Thus, the results presented in this work are important and will contribute to the parasitology community.
Response 1: We appreciate very much the comments of the first Reviewer. It was extremely gratifying to see that the reviewer captured the essence of our work, including its novelty and, at the same time, completely understood the importance of this type of contributions in eco-epidemiology research.

Reviewer 2 Report
González-Salazar et al., Toward new epidemiological landscapes of Trypanosoma cruzi transmission under future human-modified land cover and climatic change in Mexico. v.1
This is a deserving study aimed at how endemicity of Chagas disease in Mexico may be altered by the ongoing climate/environmental changes. Occurrence data on the disease and interacting species were retrieved from the “Atlas of infectious diseases of Mexico”, set against selected land cover- and BioClim covariates, and the captured relations projected into the future following three, severity-graded climate-change scenarios.
The authors - in their own terminology - “..mapped the potential distribution of T. cruzi cycles by using the Bayesian score function S(C|X(xα))..” – in the statistical literature often referred to as the “weight of evidence” (WoE). Personally, I have had a good experience with WoE applied to some vectors’ distributional data, but there is a strong concern about credibility of WoE’s predictions when the data doesn’t fully comply with statistical assumptions imposed by the model. Validation is thus highly recommended (but omitted in this study!). Presumably, the “Atlas of infectious diseases of Mexico” is a compilation of data issuing from field surveys that are scarcely uniform, rather conditioned by logistic-, security-, etc., circumstances, which typically results in oversampled central regions, and under-sampled remote provinces – a situation violating the crucial independence assumption…
Unusually, the strength of the evidence-predictor associations is in this study measured with “epsilon” [ε(Ci|Xk)] – an authors’ innovation (Ecology and Evolution. 2019;1–16). It is stated that |ε| > 1.96 corresponds to statistically significant associations (“..in he case that the binomial distribution can be approximated by a normal distribution. ..”), and serves a criterion for predictors’ selection. I don’t understand why any of the variety of established measures of association wasn’t used instead, and can identify no merit of “epsilon” - rather the opposite: I tentatively calculated “epsilon” for simulated data and found that the standards – Fisher’s exact test or chi-square - are a lot more conservative, rejecting the null at a level corresponding to “epsilon” ≈ 2.3. This evokes a suspicion that the study suffers from an excessive type I error, and that the affected conclusions (e.g. p.5, l.251, p.6, l.255, …) should be revised.
In sum, this study merits publication, but, it must be amended first. What is particularly indispensable, it is to supplement the study with validation tests. A linguistic revision is advisable.
Author Response
Response to Reviewer 2 Comments
Point 1: The authors - in their own terminology - “..mapped the potential distribution of T. cruzi cycles by using the Bayesian score function S(C|X(xα))..” – in the statistical literature often referred to as the “weight of evidence” (WoE). Personally, I have had a good experience with WoE applied to some vectors’ distributional data, but there is a strong concern about credibility of WoE’s predictions when the data doesn’t fully comply with statistical assumptions imposed by the model. Validation is thus highly recommended (but omitted in this study!). Presumably, the “Atlas of infectious diseases of Mexico” is a compilation of data issuing from field surveys that are scarcely uniform, rather conditioned by logistic-, security-, etc., circumstances, which typically results in oversampled central regions, and under-sampled remote provinces – a situation violating the crucial independence assumption…
Response 1: We agree with the reviewer’s comment. Undoubtedly, validation is an important step to build and use of geographic distribution models. In this new version we have carried out the validation of our models. For the base model (1985), the data were divided into training (70%) and testing (30%) datasets. For the projections in historical times (1993, 2002 and 2011) the validation was made using 100% of the data corresponding to each time period. In both cases, the area under the curve (AUC) of the receiver operating characteristic (ROC) was used as a validation metric.
Point 2: Unusually, the strength of the evidence-predictor associations is in this study measured with “epsilon” [ε(Ci|Xk)] – an authors’ innovation (Ecology and Evolution. 2019;1–16). It is stated that |ε| > 1.96 corresponds to statistically significant associations (“..in he case that the binomial distribution can be approximated by a normal distribution. ..”), and serves a criterion for predictors’ selection. I don’t understand why any of the variety of established measures of association wasn’t used instead, and can identify no merit of “epsilon” - rather the opposite: I tentatively calculated “epsilon” for simulated data and found that the standards – Fisher’s exact test or chi-square - are a lot more conservative, rejecting the null at a level corresponding to “epsilon” ≈ 2.3. This evokes a suspicion that the study suffers from an excessive type I error, and that the affected conclusions (e.g. p.5, l.251, p.6, l.255, …) should be revised,
Response 2: Unfortunately, epsilon is not an innovation of the authors. It is, as mentioned in the text, a binomial test and as standard as Fisher’s exact test or chi-square. As wish Fisher’s exact test, the binomial test is also exact and therefore better than an approximate test such as chi square. For choosing Fisher’s exact versus a binomial test, the latter refers to one population while the former depends on two. In terms of P(C|X) and P(C), the binomial test depends on sample size NX but considers P(C) to be exact, i.e., comes from infinite sample size, while Fisher’s exact test accounts for the fact that both NX and NC are finite. In the case that NC is small this can make a difference. In our case NC is sufficiently large that we do not believe that there is a significant difference. Unfortunately, in the referee’s simulations we do not know what sample sizes were used. Additionally, changing the threshold for epsilon from 1.96 to 2.3 will not change the large majority of our principal conclusions.
Point 3: In sum, this study merits publication, but, it must be amended first. What is particularly indispensable, it is to supplement the study with validation tests. A linguistic revision is advisable.
Response 3.
Response 3: In this new version we have included model validation and a linguistic revision.

Reviewer 3 Report
Comments on the manuscript “Toward new epidemiological landscapes of Trypanosoma cruzi transmission under future human-modified landcover and climatic change in Mexico” submitted to the Tropical Medicine and Infectious Diseases
Dear Authors
I appreciate the opportunity to evaluate this exciting manuscript.
The manuscript “Toward new epidemiological landscapes of Trypanosoma cruzi transmission under future human-modified landcover and climatic change in Mexico” deals with an ecological approach to the incidence of Chagas Disease in Mexico. It considers the climate, the presence of triatomines, the occurrence of mammals, and the environments where they live (forest, urban, and peri-residence). The authors designed three scenarios: the green, the BAU, and the worst. The analyzes were based on statistical modeling, projecting an increase in Chagas Disease in the BAU and worst-case scenarios.
It is an essential manuscript since there is a projection of scenarios for public health actions that can mitigate the effects of climate change and increase the urban niches for triatomines.
Despite the well-constructed approach to transmitting Trypanosoma sp. by triatomines in the three scenarios, I think the approach is incomplete. Triatomids are insects subjected to many pressures, which I highlight as the two most important ones. The first is bugs' vulnerability to predation. The expansion of green areas can expand the presence of bug predators; on the other hand, climate change and the devastation of the green regions, with the domiciliation of bugs, may be accompanied by an increase (or not) of predators. Most notably, arboreal mammals, (especially didelphids, edentates, and rodents), domestic avians, and some lizards are natural predators (e. g., Castello et al., 1980; Schweigmann et al., 1995a; 1995b; Lazzari et al., 2013), despite the risk of contamination. In certain circumstances, the triatomine’s predators have increased in areas with a more abundant human population, more significant numbers of houses, and decreasing green spaces (Sol et al., 2013).
The second pressure is the use of insecticides by the world population, notably in Mexico (Lopez-Carrillo et al., 1996; Liu et al., 2012; Rodríguez et al., 2018). Commonly, the use of pesticides is not directed primarily at triatomines. Generally, people use insecticides for domestic combat against the most abundant insects in homes, such as cockroaches, mosquitoes, and flies; as a side and residual effect, there is the death of many other arthropods, among which the triatomines may be. Therefore, the increased domestic use of insecticides may impact the presence of triatomines.
How do the authors consider these variables (predators and insecticides) in the model developed? Would it be possible to introduce these additional variables in estimating the incidence of Chagas Disease in Mexico? Would the lack of these variables be a weakness of the model?
It would be interesting for the authors to respond to my questions. Notwithstanding, I consider the manuscript to be well structured and well-written.
References
Castello J. A. & Gil Rivas M. J. (1980). Propuesta de un predador para la destrucción de la vinchuca: la salamanquesa común (Tarentola mauritanica) [Proposal for a predator for the destruction of Triatoma infestans: Tarentola mauritanica]. Medicina (B Aires).40:673-7. Spanish. PMID: 22167700.
Lazzari, C. R., Pereira, M. H., & Lorenzo, M. G. (2013). Behavioural biology of Chagas disease vectors. Memorias do Instituto Oswaldo Cruz, 108, 34-47.
López-Carrillo, L., Torres-Arreola, L., Torres-Sánchez, L., Espinosa-Torres, F., Jiménez, C., Cebrián, M., ... & Saldate, O. (1996). Is DDT use a public health problem in Mexico?. Environmental Health Perspectives, 104(6), 584-588.
Liu, Y., Liu, F., Pan, X., & Li, J. (2012). Protecting the environment and public health from pesticides.
Rodríguez, A. G. P., López, M. I. R., Casillas, Á. D., León, J. A. A., & Banik, S. D. (2018). Impact of pesticides in karst groundwater. Review of recent trends in Yucatan, Mexico. Groundwater for Sustainable Development, 7, 20-29.
Schweigmann, N. J., Pietrokovsky, S., Conti, O., Bottazzi, V., Canale, D. & Wisnivesky-Colli, C. (1995a). The interaction between poultry and Triatoma infestans Klug, 1834 (Hemiptera: Reduviidae) in an experimental model. Memórias do Instituto Oswaldo Cruz, 90, 429-431.
Schweigmann, N. J., Pietrokovsky, S., Conti, O., Bottazzi, V., Canale, D. & Wisnivesky-Colli, C. (1995b). The interaction between poultry and Triatoma infestans Klug, 1834 (Hemiptera: Reduviidae) in an experimental model. Memórias do Instituto Oswaldo Cruz, 90, 429-431.
Sol, D.; Lapiedra, O.; González-Lagos, C. (2013). Behavioural adjustments for a life in the city. Animal Behaviour, 85(5), 1101–1112.
Author Response
Response to Reviewer 3 Comments
Dear Authors
I appreciate the opportunity to evaluate this exciting manuscript.
The manuscript “Toward new epidemiological landscapes of Trypanosoma cruzi transmission under future human-modified landcover and climatic change in Mexico” deals with an ecological approach to the incidence of Chagas Disease in Mexico. It considers the climate, the presence of triatomines, the occurrence of mammals, and the environments where they live (forest, urban, and peri-residence). The authors designed three scenarios: the green, the BAU, and the worst. The analyzes were based on statistical modeling, projecting an increase in Chagas Disease in the BAU and worst-case scenarios.
It is an essential manuscript since there is a projection of scenarios for public health actions that can mitigate the effects of climate change and increase the urban niches for triatomines.
Point 1: Despite the well-constructed approach to transmitting Trypanosoma sp. by triatomines in the three scenarios, I think the approach is incomplete. Triatomids are insects subjected to many pressures, which I highlight as the two most important ones. The first is bugs' vulnerability to predation. The expansion of green areas can expand the presence of bug predators; on the other hand, climate change and the devastation of the green regions, with the domiciliation of bugs, may be accompanied by an increase (or not) of predators. Most notably, arboreal mammals, (especially didelphids, edentates, and rodents), domestic avians, and some lizards are natural predators (e. g., Castello et al., 1980; Schweigmann et al., 1995a; 1995b; Lazzari et al., 2013), despite the risk of contamination. In certain circumstances, the triatomine’s predators have increased in areas with a more abundant human population, more significant numbers of houses, and decreasing green spaces (Sol et al., 2013).
How do the authors consider these variables (predators and insecticides) in the model developed? Would it be possible to introduce these additional variables in estimating the incidence of Chagas Disease in Mexico? Would the lack of these variables be a weakness of the model?
Response 1: Reviewer point an important topic on the role of potential triatomine´s predators on incidence of T. cruzi. Without a doubt, vertebrate predators may impact triatomine populations, but at the same time, predation can also be a route to T. cruzi infection. We agree with the reviewer’s comment that human-modified landscapes can expose triatomines to predators, however, contrasting results have been observed evaluating the role of domestic species on triatomine incidence. On the one hand, domestic birds, such as ducks, can be important predators of triatomines (Schweigmann et al., 1995), on the other hand, chickens can favour the infestation of some triatomines in rural houses (Cecere et al., 1997). In addition, evaluation of triatomine predation has been restricted to only a few regions. Consequently, the scarcity of this type of information does not allow us to integrate it into our model, however, if these data were available, our methodology is capable of integrating them. Although we do not consider triatomine predation data, this is not a weakness of our model because our analysis is not based only on the presence of triatomine species, but also integrates information on the presence of vertebrate hosts that participate in the transmission and maintenance of T. cruzi transmission cycles.
Cecere, M. C., Gürtler, R. E., Chuit, R., & Cohen, J. E. (1997). Effects of chickens on the prevalence of infestation and population density of Triatoma infestans in rural houses of north‐west Argentina. Medical and Veterinary Entomology, 11(4), 383-388.
Point 2: The second pressure is the use of insecticides by the world population, notably in Mexico (Lopez-Carrillo et al., 1996; Liu et al., 2012; Rodríguez et al., 2018). Commonly, the use of pesticides is not directed primarily at triatomines. Generally, people use insecticides for domestic combat against the most abundant insects in homes, such as cockroaches, mosquitoes, and flies; as a side and residual effect, there is the death of many other arthropods, among which the triatomines may be. Therefore, the increased domestic use of insecticides may impact the presence of triatomines.
How do the authors consider these variables (predators and insecticides) in the model developed? Would it be possible to introduce these additional variables in estimating the incidence of Chagas Disease in Mexico? Would the lack of these variables be a weakness of the model?
Response 2: Control of vector species has been based on the use of insecticides, which is a potential source of pressure on triatomine populations. However, the emergence of insecticide-resistance for some triatomine species has been observed (e.g., Pedrini et al., 2009, Mougabure-Cueto & Picollo, 2015). In addition, the re-infestation of triatomines after a few months of insecticide application (e.g., Gorla, 1991; Dumonteil et al.., 2004) has been documented. Although the efficacy of the use of insecticides as a triatomine control measure is not clear, it would be interesting to include it in the model. However, there is no geographic information available on insecticide application plans. However, not including this information does not represent a weakness in our model. As mentioned before, we include both vectors and hosts.
Dumonteil, E., Ruiz-Piña, H., Rodriguez-Félix, E., Barrera-Pérez, M., Ramirez-Sierra, M. J., Rabinovich, J. E., & Menu, F. (2004). Re-infestation of houses by Triatoma dimidiata after intra-domicile insecticide application in the Yucatan peninsula, Mexico. Memórias do Instituto Oswaldo Cruz, 99, 253-256.
Gorla, D. E. (1991). Recovery of Triatoma infestans populations after insecticide application: an experimental field study. Medical and Veterinary Entomology, 5(3), 311-324.
Mougabure-Cueto, G., & Picollo, M. I. (2015). Insecticide resistance in vector Chagas disease: evolution, mechanisms and management. Acta Tropica, 149, 70-85.
Pedrini N, Mijailovsky SJ, Girotti JR, Stariolo R, Cardozo RM, Gentile A, et al. (2009) Control of Pyrethroid-Resistant Chagas Disease Vectors with Entomopathogenic Fungi. PLoS Negl Trop Dis 3(5): e434. https://doi.org/10.1371/journal.pntd.0000434.

Round 2
Reviewer 2 Report
González-Salazar et al., Toward new epidemiological landscapes of Trypanosoma cruzi transmission under future human-modified land cover and climatic change in Mexico. v.2
I acknowledge that the manuscript has been amended substantially, and that it can be – conditionally on a minor revision detailed below – recommended for publication.
P.1, l.21: “important neglected disease” sounds strange - what about “important yet neglected…”?
P.1, l.31: insert comma before “historical”, pls.
P.1, l.32 and throughout the article: I recommend the authors to consistently use the unabridged term “transmission cycle” as, at places, it isn’t clear whether the parasite’s life cycle or its zoonotic transmission cycle is concerned
P.1, l.66: lowercase “orders”, pls.
P.3., l.100: a dynamics
P.3, l.103: “commensal pests” can’t be counted among domesticated animals – I suggest “..domestic mammals (dogs, cats) and synanthropes, such as rats,..”
P.3, l.131: delete “First,” pls, (there is no “second” counterpart...)
P.3, l.132: information on
P.3, l.133-6: this sentence is neither coherent nor essential for understanding, and could be left out completely (it only replays what is said in M&M)
P.4, l.170: “annual precipitation” is the correct name for Bio12 in the WorldClim documentation..
P.4, l.183-4 and throughout: substitute “presence/absence” for “presence/no presence” - just to be in line with the mainstream terminology..
P.4, l. 185-9: concurrently with explaining this referee in the cover letter that “..epsilon is not an innovation of the authors..”, it is “..as standard as Fisher’s exact test..”, etc., etc., the authors should – first and foremost - preclude the reader from such a misunderstanding - best by steering clear of referring to their own work in this context, and substituting some pertinent literature source for the self-citation ([35])…
P.5, l.206-10: “..we first grouped T. cruzi cycle presences..” – presences only, or both presences and absences? Make it unambiguous, pls.
P.5, l.225: “…we randomly split the presence data…” – ditto
P.5, l.252-3: “…> 70% of mammals have been observed in both natural and anthropogenic habitats and are thus synanthropic species” - mere finding of a species in human proximity doesn’t make it a synanthrope. By definition, a characteristic feature of a synanthrope is that it benefits from the coexistence with humans. Many species simply have no choice of an alternative habitat. Revise it, pls.
P.5, l.253-p.6, l.255: “..Records of domestic species that are positive for T. cruzi include domestic pets (dogs and cats), livestock species (e.g., cows, horses, pigs) and exotic rodents (rats and mice) ..” – what do the authors mean with “exotic rodents”? In Table S1, there are listed no rodent species kept as pets (except for ‘noble breeds’ of the domestic mouse and the brown rat – i.e. species classified here as “commensal pests”). Further in the text it is written that: “..Dogs and exotic rodents have been recorded in both wild and domestic habitats..”, which suggests that “exotic rodents”, in fact, equal “commensal pests”. The authors should be more careful in manipulating with terms on the borderline between domestic, synanthropic, pest, etc.
P.6, l.260: the pathogen’s transmission
P.6, l.276-7: throughout 1985 - 2002 ?
P.7, l.283-4: “..there has been a domiciliation process in urban populations ..” – populations of what? T.cruzi or bugs? Be specific, pls.
P.8, l.324: LUCC- and climate changes
P.11, l.351-2: an important yet neglected disease
P.11,l.361-2: “..Consequently, there is no link between triatomine distribution and T. cruzi presence, and between the influence of changes in land cover and climate. ..” – at least some reference supporting these claims is desirable..
P.12, l.381: the role of climatic conditions in
P.12, l.382: in recent time-horizons
Author Response
Response to Reviewer 2 Comments
We thank the Reviewer for their detailed report and relevant criticisms. We have reviewed our manuscript paying careful attention to each of them. We address their detailed comments below:
Point 1: P.1, l.21: “important neglected disease” sounds strange - what about “important yet neglected…”?
Response 1: We have added “yet”
Point 2: P.1, l.31: insert comma before “historical”, pls.
Response 2: We have adjusted the text accordingly
Point 3: P.1, l.32 and throughout the article: I recommend the authors to consistently use the unabridged term “transmission cycle” as, at places, it isn’t clear whether the parasite’s life cycle or its zoonotic transmission cycle is concerned
Response 3: We have adjusted the text accordingly
Point 4: P.1, l.66: lowercase “orders”, pls.
Response 4: We have changed this
Point 5: P.3., l.100: a dynamics
Response 5: We have changed this
Point 6: P.3, l.103: “commensal pests” can’t be counted among domesticated animals – I suggest “..domestic mammals (dogs, cats) and synanthropes, such as rats,..”
Response 6: We have adjusted the text accordingly
Point 7: P.3, l.131: delete “First,” pls, (there is no “second” counterpart...)
Response 7: We have deleted “First”
Point 8: P.3, l.132: information on
Response 8: We have changed this
Point 9: P.3, l.133-6: this sentence is neither coherent nor essential for understanding, and could be left out completely (it only replays what is said in M&M)
Response 9: We have deleted this sentence
Point 10: P.4, l.170: “annual precipitation” is the correct name for Bio12 in the WorldClim documentation..
Response 10: We have changed this
Point 11: P.4, l.183-4 and throughout: substitute “presence/absence” for “presence/no presence” - just to be in line with the mainstream terminology..
Response 11: We have adjusted the text accordingly
Point 12: P.4, l. 185-9: concurrently with explaining this referee in the cover letter that “..epsilon is not an innovation of the authors..”, it is “..as standard as Fisher’s exact test..”, etc., etc., the authors should – first and foremost - preclude the reader from such a misunderstanding - best by steering clear of referring to their own work in this context, and substituting some pertinent literature source for the self-citation ([35])…
Response 12: We have included several such references
Point 13: P.5, l.206-10: “..we first grouped T. cruzi cycle presences..” – presences only, or both presences and absences? Make it unambiguous, pls.
Response 13: We used presence only, we have adjusted the text accordingly
Point 14: P.5, l.225: “…we randomly split the presence data…” – ditto
Response 14: We used presence only, we have adjusted the text accordingly
Point 15: P.5, l.252-3: “…> 70% of mammals have been observed in both natural and anthropogenic habitats and are thus synanthropic species” - mere finding of a species in human proximity doesn’t make it a synanthrope. By definition, a characteristic feature of a synanthrope is that it benefits from the coexistence with humans. Many species simply have no choice of an alternative habitat. Revise it, pls.
Response 15: We agree with the reviewer comment, we have deleted the term “synanthropic species” and adjusted the text accordingly
Point 16: P.5, l.253-p.6, l.255: “..Records of domestic species that are positive for T. cruzi include domestic pets (dogs and cats), livestock species (e.g., cows, horses, pigs) and exotic rodents (rats and mice) ..” – what do the authors mean with “exotic rodents”? In Table S1, there are listed no rodent species kept as pets (except for ‘noble breeds’ of the domestic mouse and the brown rat – i.e. species classified here as “commensal pests”). Further in the text it is written that: “..Dogs and exotic rodents have been recorded in both wild and domestic habitats..”, which suggests that “exotic rodents”, in fact, equal “commensal pests”. The authors should be more careful in manipulating with terms on the borderline between domestic, synanthropic, pest, etc.
Response 16: We have adjusted the text to avoid confusion with the terms domestic, exotic and synanthropic
Point 17: P.6, l.260: the pathogen’s transmission
Response 17: We have adjusted the text accordingly
Point 18: P.6, l.276-7: throughout 1985 - 2002 ?
Response 18: Correlation was observed in 1985, 1993 and 2002, We have adjusted the text accordingly
Point 19: P.7, l.283-4: “..there has been a domiciliation process in urban populations ..” – populations of what? T.cruzi or bugs? Be specific, pls.
Response 19: We refer to the domiciliation of bugs, we have adjusted the text accordingly
Point 20: P.8, l.324: LUCC- and climate changes
Response 20: We have added “changes”
Point 21: P.11, l.351-2: an important yet neglected disease
Response 21: We have added “yet”
Point 22: P.11,l.361-2: “..Consequently, there is no link between triatomine distribution and T. cruzi presence, and between the influence of changes in land cover and climate. ..” – at least some reference supporting these claims is desirable..
Response 22: We have included several such references
Point 23: P.12, l.381: the role of climatic conditions in
Response23: We have changed this
Point 24: P.12, l.382: in recent time-horizons
Response 24: We have changed this
